# Risk Factors, Radiological and Clinical Outcomes in Subclinical and Clinical Pituitary Apoplexy

**DOI:** 10.3390/jcm11247288

**Published:** 2022-12-08

**Authors:** Betina Biagetti, Silvana Sarria-Estrada, Esteban Cordero Asanza, Anas Chaachou-Charradi, Yiken Karelys Ng-Wong, Marta Cicuendez, Irene Hernandez, Alba Rojano-Toimil, Pilar Costa, Elena Martinez-Saez, Anna Casteràs, Rafael Simò

**Affiliations:** 1Endocrinology Department, Vall d’Hebron University Hospital, 08035 Barcelona, Spain; 2Diabetes and Metabolism Research Unit, Vall d’Hebron Research Institute (VHIR), 08035 Barcelona, Spain; 3Universidad Autónoma de Barcelona, 08193 Barcelona, Spain; 4Neuroradiology Section, Radiology Department, Vall d’Hebron University Hospital, 08035 Barcelona, Spain; 5Neurosurgery and Neurotraumatology, Vall d’Hebron University Hospital, 08035 Barcelona, Spain; 6Pathology Department, Vall d’Hebron University Hospital, 08035 Barcelona, Spain

**Keywords:** pituitary apoplexy, pituitary adenoma, hemorrhage, necrosis, neurosurgery, emergency, subclinical apoplexy

## Abstract

Background: Pituitary apoplexy (PA) can be symptomatic, namely acute apoplexy (APA), or asymptomatic or subclinical (SPA). Objective: To describe the clinical characteristics and evolution of the patients with APA compared to SPA Patients and methods: Retrospective, longitudinal database analysis. Results: We identified 58 patients with PA, and 37 accomplished the inclusion criteria (17 men, median age 47.7 years). A total of 29 (78.4%) had APA (17 underwent surgery, and 12 were conservatively managed), and 8 (21.6%) had SPA. The presence of non-functioning pituitary adenoma (NFPA) odds ratio (OR): 29.36 (95% confidence interval (CI): 1.86–462.36) and the largest size OR 1.10 (95% CI: 1.01–1.2) elevated the risk of having surgery. Hypopituitarism developed in 35.1% without significant differences between APA and SPA. In non-surgical patients, adenoma volume shrunk spontaneously at one year magnetic resonance imaging (MRI), without statistical differences between the conservatively treated and SPA group. Conclusions: APA is more frequent in larger NFPAs, and this subset of patients has a higher risk of surgery. Hypopituitarism is quite frequent even in patients with SPA, and, therefore, long-term follow-up is mandatory. In the non-surgical group, the pituitary tumour shrinkage is clinically relevant after one year of PA. Consequently, surgery indication in NFPA should be delayed and reassessed if patients remain asymptomatic.

## 1. Introduction

Pituitary apoplexy (PA) is a rare, sometimes life-threatening clinical syndrome as a result of acute/subacute haemorrhage and/or infarction of a pituitary tumour, and a severe sudden onset headache is one of its main symptoms [1,2]. Nevertheless, some subclinical PA (SPA) are incidentally discovered in the follow-up of a known pituitary adenoma or when magnetic resonance imaging (MRI) is performed for other conditions. The frequency of SPA has been estimated according to different studies in 10–25% of patients with pituitary adenoma [1,2,3,4], whereas symptomatic acute pituitary apoplexy (APA) has been reported less frequently (2–10% of the patients) [1,4]. Nevertheless, the wide spectrum of clinical presentation and the lack of agreement on the clinical definition of the entity make its prevalence and incidence difficult to estimate.

Traditionally, PA has been described as more frequent in non-functioning pituitary adenomas (NFPAs) [4,5,6]. This is logical because NFPAs are one of the most frequent pituitary tumours and generally are larger than functional adenomas. In addition, when emergency surgery is undertaken, in many cases, there is a lack of well-established hormone evaluation, and the pathology origin is impossible to assess due to the extent of necrosis.

In less than 50% of cases, there are some precipitating factors reported in the literature, such as prior major surgery (mainly orthopaedic and cardiac surgery), brain traumatism, endocrinological dynamic test, and cerebral angiographic procedures, as well as some drugs (e.g., antiplatelet, anticoagulants, dopamine agonist, or gonadotrophin agonist). In addition, PA is less frequent and has a more favourable prognosis in the setting of microadenoma [7]. Moreover, one large surgical series, including 109 patients with PA, suggests that the clinical picture of PA could be graded according to the clinical presentation from subclinical to severe symptomatic with acute visual deficits or decreased Glasgow scale [8]. However, this information could be biased towards APA because is based on hospitalised patients or surgical series.

Regarding treatment, traditionally, PA has been universally considered a neurosurgical emergency [9]. In fact, this has been extrapolated from intracranial hematoma management, where early relief of mass effect has demonstrated superior clinical outcomes in the setting of brain compression [10,11]. Nowadays, high doses of glucocorticoid are mandatory immediately after PA diagnosis, and neurosurgery is reserved for more severe cases or when the conservative approach fails [11,12,13,14,15]. However, the timing of surgery for improving visual outcomes is a matter of debate [16,17].

Finally, little is known regarding pituitary function and tumour volume evolution after apoplexy.

In summary, pituitary apoplexy remains a poorly understood condition [18]. Potential risk factors are eventually biased because the main series are based on hospitalised patients or surgical registries. In addition, the outcomes regarding pituitary function and volume evolution after PA are not well-known. In the present study, a multidisciplinary experience comparing clinical characteristics and evolution of surgical and non-surgical patients with APA and SPA apoplexy is reported. Our results will help to better understanding and improve the management of this complication of pituitary tumours.

## 2. Patients and Methods

We analysed our database searching for patients diagnosed with PA from 2010 to 2021. This registry is annually updated in compliance with CSUR (Reference Centres, Services and Units within the Spanish Health System (SNS)) and Endo-ERN (Endocrine European Reference Network) designation. Records are obtained based on electronic codes regarding the following items: (1) Emergency room’ or ICU diagnostic; (2) Ward admission diagnosis; (3) Surgical diagnosis; (4) Pathology diagnosis; (5) Medical consultation diagnosis. We followed the Strengthening the reporting of observational studies in epidemiology (STROBE) Statement for observational studies [19]


Definitions:


Subclinical PA (SPA): presence of radiological evidence of haemorrhage or necrosis in the pituitary gland incidentally discovered (absence of symptoms).

Acute apoplexy (APA): patient consulting for symptoms consistent with PA (headache, visual disturbances, and/or altered conscientiousness) with radiological confirmation of PA.

Figure 1 schematically shows APA and SPA in macro and microadenomas.

Conservative management: When patients did not require surgery, or surgery was delayed and scheduled after one month. By protocol, we used a bolus of hydrocortisone (100 mg intravenous) followed by 50 mg every 6 h iv. in any patient with PA suspicion while waiting for an ophthalmologist and neurosurgery evaluation.

Surgical management: It was considered when the patient required surgery within 1 week after PA diagnosis. All the patients had an endonasal endoscopic transsphenoidal surgical approach for tumour removal, and the sample was sent for histopathological examination.

Fail to conservative management: Patients who required surgery after 1 week of conservative approach and before one month from the APA due to the absence of clinical improvement.

The study was conducted according to the mandates in the Declaration of Helsinki, and the Ethics Committees of the Vall d’Hebron University Hospital approved all procedures. All neuroimages were reviewed by two neuroradiologist (SS and YKN).


Inclusion criteria


Inclusions criteria were: (1) patients older than 18 years at apoplexy diagnosis with a follow-up conducted in our Hospital; (2) imaging confirmation (MRI scanning of the pituitary with haemorrhage or necrosis); (3) at least one MRI at one year of follow-up, except in case of death.


Variables and measurements


Clinical characteristics: age, gender, body mass index, history of diabetes, hypertension, dyslipidaemia, smoking habit, cardiovascular disease, hypopituitarism, haematological disease, cancer, general surgery, pregnancy, and dynamic endocrinological tests. Drug abuse and medications, including contraceptives, antiplatelets, or anticoagulants, were also recorded. 

In the group of APA: signs, symptoms, number of days with symptoms, treatment options, date of surgery and eventual complications, corticosteroid doses, and pathology results were recorded. In all patients, the following outcomes from the first day of PA to the end of follow-up were recorded: hyponatremia, neurological and hormonal status, tumour changes, and death.

Tumour characteristics: Size, parasellar tumour involvement by Knosp grading [20], apoplexy characteristics including haemorrhage, necrosis, and signal intensities on T1 and T2-weighted images on MRI.

Pituitary MRI: The standard MRI (3T magnets, Siemens) protocol included non-enhanced and contrast-enhanced coronal and sagittal 2D T1 weighted image (T1-WI), and coronal 2D T2 weighted image (T2-WI) (slice thickness 3 mm). The maximum tumour diameter was measured and used to classify pituitary adenoma as microadenoma (<10 mm) or macroadenomas (≥10 mm). The percentage of tumour shrinkage was calculated using the two largest diameters in sagittal view as previously reported [(%) = (Basal D1 × D2 − Post D1 × D2) × 100/Basal DI × D2] [21].


Statistical analysis


Categorical variables are expressed as a percentage and compared using the χ^2^ test or with Fisher’s exact test when at least one of the expected frequencies was less than 5. Continuous variables are expressed as the median and interquartile range (IQR), and comparison between two groups was performed using Student’s *t*-test or the nonparametric Mann–Whitney U-test accordingly to variable distribution.

We used Spearman’s rank test to investigate which variables among demographic variables, past history, or tumour characteristics influenced to have surgery. Predictors with *p* < 0.20 were included in multiple logistic regression, and a stepwise backward selection approach was used to identify the best predictive model for surgery. The model fit and calibration were assessed by Akaike’s and Bayesian information criterion, the Hosmer–Lemeshow goodness-of-fit test, and the discrimination power by calculating the area under the receiver operating characteristic (ROC) curves (AUC). Results were expressed as odds ratio (OR) and 95% CI. A *p*-value of <0.05 was considered significant. Statistical analyses were performed using the STATA16 statistical package (College Station, TX 77845 USA) for Windows.

## 3. Results

From February 2010 to August 2021, we identified 58 patients with PA of a total of 529 patients with pituitary lesions (121 NFPA, 189 prolactinomas, 74 acromegalies, 68 Cushing disease, and 77 craniopharyngiomas). Of this, 37 accomplished the inclusion criteria. A total of 21 were excluded for the following reasons: eleven because the first 2–3 days were managed in other hospitals, thus impeding our full access to data (clinical presentation, treatment, and/or imaging); eight SPA because a second MRI was lacking and/or the follow-up, was not conducted in our Hospital and two cases because after a careful evaluation by the radiologists the lesion was defined as a Rathke cleft cyst (Figure 2). Overall, 37 patients met the inclusion criteria for this study and were analysed.

### 3.1. Main Differences between SPA and APA

The clinical features of the total included patients, patients with SPA, and APA are summarised in (Table 1). The majority of patients (29, 78.4%) had APA, whereas 8 (21.6%) had SPA. There was no statistical difference between patients with APA and those with SPA except for Type 2 Diabetes Mellitus (T2DM) and dyslipidemia, which were more frequent in the APA group (37.9% vs. 0%; *p* < 0.04, and 34% vs. 0%; *p* < 0.05, respectively).

Regarding tumour characteristics: 13 (35.1%) were previously diagnosed with pituitary adenoma (8 NFPAs, 4 prolactinomas, and 1 acromegaly), of which only 3 were microadenomas (2 NFPAs and 1 prolactinoma). The median time from the diagnosis of pituitary adenoma to PA was 27.3 (IQR 59.3) months. Of the remaining 24 patients without prior know adenoma, the majority were non-functional macroadenomas (16, 66.7%), followed by prolactinomas (6, 25.0%) and GH-secreting tumours (2, 8.3%). As a whole, there were 24 (64.9%) NFPAs, 10 (27.0%) prolactinomas, and 3 (8.1%) GH-secreting adenomas. The prevalence of apoplexy in each tumour type in our series was (24/121, (19.8%), for NFPA), (10/189, (5.2%) for prolactinomas) and (3/74 (4.1%) for acromegaly). We did not find any PA in patients with Cushing disease.

Patients with SPA had smaller (13.5 vs. 26.0 mm; *p* < 0.01) and less invasive tumours (2, 25.0% vs. 25, 86.2% *p* < 0.01) than those patients who suffered APA, and all of them were hyperintense in T1-WI (Table 2).

Hyponatremia was present in 7 out of 37 (18.9%) cases, 6 (85.7%) had adrenal insufficiency, and 1 (14.3%) presented a SIADH, all of them belonging to the APA group.

### 3.2. Differences between Patients with Conservative vs. Surgical Approach in APA

Clinical presentation and tumour characteristics and the type of treatment (conservative or surgical) are displayed in Table 3. Headache (26, 89.7%) and visual impairment (21, 72.4%) were the most frequent manifestations. There were no statistical differences between groups in clinical presentation except for photophobia, which was more frequent in the surgery group (8 (47.0%) vs. 1 (8.3%); *p* < 0.03). The surgery group had larger tumours (30.5 ± 9.0 mm vs. 20.5 ± 11.5; *p* < 0.03), with more frequent suprasellar extension and optic chiasm compression (16, 94.1% vs. 7, 58.3%; *p* < 0.03) than the group treated with conservative measures.

All patients with APA received glucocorticoid treatment without statistical differences in the doses employed in the first 24 h between those treated with conservative measures and those requiring surgery.

### 3.3. Clinical and Radiological Outcomes

All the patients under conservative management improved clinical signs and symptoms except three patients who required surgery. The development of hypopituitarism during follow-up considering the presence of SPA vs. APA and the type of treatment is shown in Figure 3. We did not find any statistical difference between SPA and APA. In SPA, 1 (12.5%) patient had hypopituitarism at diagnosis and 2 (25.0%) in the follow-up. In the case of the APA group, the patients that followed the conservative approach ((3, 25%) and (4, 33%)) and in the surgery group ((2, 11.8%) and (7, 41.2%)) had hypopituitarism at diagnosis and in the follow-up, respectively.

The baseline and changes in pituitary tumour volume at 12 months of follow-up after PA in patients in whom surgery was not required are displayed in Figure 4. The adenoma shrunk after PA in 13 out of 20 (65%) of non-surgical cases, with a median percentage of volume reduction at 1 year of 23.7% in the SPA group and 40.0% in APA managed by conservative measures.

In the follow-up, four patients died, but only one was related to PA. A new apoplexy episode was observed in two cases.

### 3.4. Pathology

In 3 out of 17 surgical cases, the pathology evaluation was not possible due to massive haemorrhage or necrosis. In the remaining 14 cases, we did not find any specific characteristic that could identify a more aggressive behaviour of these tumours. Only in one case was the Ki67 near 10%, but there was no tumour regrowth or relapse after five years of follow-up.

### 3.5. Prediction Model for Surgery:

We used Spearman’s rank test and stepwise backward selection approach to identify the best predictive model for surgery. The included variables were maximum diameter of the pituitary tumour, age, gender, body mass index, hyper T1 and T2 signalling, presence of T2DM, dyslipidaemia, smoking habit, cranial nerve involvement, chiasma compression, and presence of NFPA (extrasellar and sinus invasion were excluded due to co-linearity). The reduced and parsimonious model that best explained the probability of required surgery included: having a NFPA, hyperintensity on T1-WI, and maximum diameter on MRI. The model was well-fitted according to Akaike’s and Bayesian information criteria, and the likelihood ratio compared with the null model was <0.01. The binomial area under the receiver operating characteristic curve (AUC) was 0.89 CI (0.73–0.98). The Youden index was 0.70 (sensibility: 82.4%; specificity: 85.0%; correct classification: 83.8%) (Figure 5 and Table 4).

## 4. Discussion

In our series, patients with APA had more frequent larger and invasive NFPAs, as well as T2DM and dyslipidemia. Conversely, in SPA, the tumours were smaller and less invasive, and characteristically all of them had hyperintense in T1-WI. Despite these differences, it is remarkable that in the follow-up, the hypopituitarism, as well as a clinically relevant tumour volume reduction at one year of the follow-up, was similar in both groups.

Our results are in line with the proposed grading apoplexy scale by Jho DH et al. [8], in which grade 1 corresponded to asymptomatic patients (those with SPA), and grade 5 were patients with visual acuity/field deficits or decreased level of consciousness. Patients with smaller tumours tended to have SPA, and the bigger and more invasive were symptomatic. Similarly, Nakhleh et al. [7] studied 27 patients with pituitary apoplexy, of which 7 (26%) had microadenomas, and these cases presented a more favourable prognosis.

In our series, up to 38% of the patients with APA had T2DM, and none of the patients with SPA. Diabetic microangiopathy, haemodynamic changes, and the pro-thrombotic state that occurs in patients with uncontrolled diabetes could favour pituitary apoplexy [22,23,24]. However, although plausible, some works have not found this association in acute apoplexy [25,26] or subclinical apoplexy [27].

Similarly to the results observed in the present study, Singh et al. [28] found that up to 35.1% of the patients had a previously known adenoma without differences in SPA vs. APA. The time to develop the apoplectic event was highly variable, and therefore, it was not possible to establish a cut-off time from which the apoplectic event could be more or less frequent in patients with a known pituitary tumour.

The most common tumour in our PA series was NFPA. These results are consistent with other series [2,4,25]. The susceptibility of NFPA to apoplexy deserves special mention. McCabe et al. [29] reported a relationship between the expression of vascular endothelial growth factor (VEGF) and tumour growth. However, other groups failed to show this association [30,31]. Recently, Akbari et al. [32] found increased expression of cyclooxygenases type 2 (COX-2) in NFPA compared with the other pituitary tumours. It should be noted that cyclooxygenases are related to pathways mediating vasodilation, platelet aggregation, and inflammation, among others. Furthermore, the COX-2 expression level was significantly increased in macroadenomas and invasive tumours. Accordingly, VEGF and COX-2, among others, could be behind the susceptibility of NFPA to bleeding or infarction. Other pathways such as tumour necrosis factor α, matrix metalloproteinases [33], hypoxia-inducible factor 1 [34], and pituitary tumour transforming gene [35] have also been linked with PA [18].

In the clinical setting, Semple et al. [36] showed an accurate correlation between MRI findings and histological pattern in PA. They studied the MRI, histology, and clinical outcomes in 38 patients with APA and found that the patients who presented infarction alone had less severe clinical features and better outcomes than those who had haemorrhage alone or combined with infarction. In our series, MRI revealed that the majority of patients had mixed components of haemorrhage and infarction. Due to these components, and especially for the usual pattern of blood degradation, the signal on both T1-WI and T2-WI usually are heterogeneous (the signal varies over time due to changes in deoxyhaemoglobin, methaemoglobin, and haemosiderin/ferritine). However, in contrast to APA, all the patients with SPA in our series had hyperintensity signals on T1-WI, and half of them also had hyperintensity on T2-WI. In this regard, it should be noted that the degradation of blood in the subacute phase (methaemoglobin) is characterised by hyperintensity on T1-WI (28). Therefore, the hyperintensity signal on T1-WI observed in SPA could be because SPA is asymptomatic, and we diagnosed it later on in non-acute phases. Thus, radiological evaluation could guide one chronological aspect and can help to make clinical decisions.

To our knowledge, this is the first study to assess and compare the clinical course and long-term outcomes of apoplexy in SPA vs. APA. Liu et al. [6] evaluated 65 patients after trans-sphenoidal surgery who had radiological and pathological confirmation of pituitary haemorrhage. Although they divided patients into clinical or subclinical PA, those with subclinical apoplexy presented headache (30%) and visual deficits (70%). Thus, this study actually compared mild symptomatic vs. acute apoplexy in patients requiring surgery and, therefore, can hardly be compared with what herein presented. Kinoshita et al. [4] reviewed 58 patients with PA of 328 consecutive patients who underwent surgery for pituitary adenoma. SPA defined as an intra-tumoral haemorrhage based on MRI and surgical findings, and absence of clinical symptoms of apoplexy was detected in 47, and 11 were classified as APA. They investigated the pituitary dysfunction 3 months after surgery using pre and postoperative provocative tests in 54 of the 58 patients with PA. They found that SPA did not cause any dysfunction in pituitary glands after surgery. These results are in disagreement with our findings showing a similar rate of hypopituitarism in APA and SPA. A possible explanation is that the follow-up period of our study was at least 23.0 ± 42.0 months and pituitary function was not only assessed 3 months after surgery. This point out that long-term hormonal follow-up should be mandatory in patients with PA, even in patients with SPA.

In addition, we found that in non-surgical patients, the tumour volume shrunk after apoplexy in 65% of the patients, and none of them re-grew. On the other hand, two patients with NFPA in the APA group had new apoplexy in the follow-up, thus re-enforcing the concept that a predisposition to bleeding could be present in NFPA.

Regarding the treatment of choice in APA, the modalities of treatment and times of intervention have been largely debated. The risk–benefit ratio of conservative treatment vs. surgery should be carefully evaluated. Singh et al. [28] described the outcomes of 87 patients with PA at a single institution. In this series, non-statistical differences in outcomes were found between patients who underwent surgery or were conservatively treated. A meta-analysis including 200 patients with PA aimed at evaluating visual recovery comparing early vs. late surgery did not find statistical differences [37]. Likewise, other studies found similar pituitary impairment and significant improvement in outcomes regardless of the timing of surgical intervention [16,17,38,39,40]. The ongoing multicenter project conducted at Cedars-Sinai Hospital called “Pituitary Apoplexy Surgical Timing and Outcome Registry (PASTOR)” would l provide definitive evidence about treatment approaches and outcomes [41].

Finally, in our series, only three patients failed conservatory treatment and required surgery, but one of them, in fact, had clinical criteria for acute surgery but was delayed due to active oncological treatment.

The prediction model for undergoing surgery confirmed that the risk of having a PA and requiring surgery was many times higher in NFPA and larger tumours.

Our study has some limitations, such as the retrospective design in a single centre. The sample size did not allow the analysis of events with low frequency. In addition, being a referral hospital could imply a potential bias towards more APA than SPA. The main strengths were the rigorous selection and good characterisation of the patients included in the study.

In conclusion, these data suggest that larger NFPA adenomas have a higher risk of APA and could require surgery. Hypopituitarism is quite frequent even in patients with SPA, and therefore, long-term follow-up is mandatory. In the non-surgical group, the pituitary tumour shrinkage is clinically relevant after one year of PA, and consequently, surgical options should be delayed and reassessed if the patients remain asymptomatic.

## Figures and Tables

**Figure 1 jcm-11-07288-f001:**
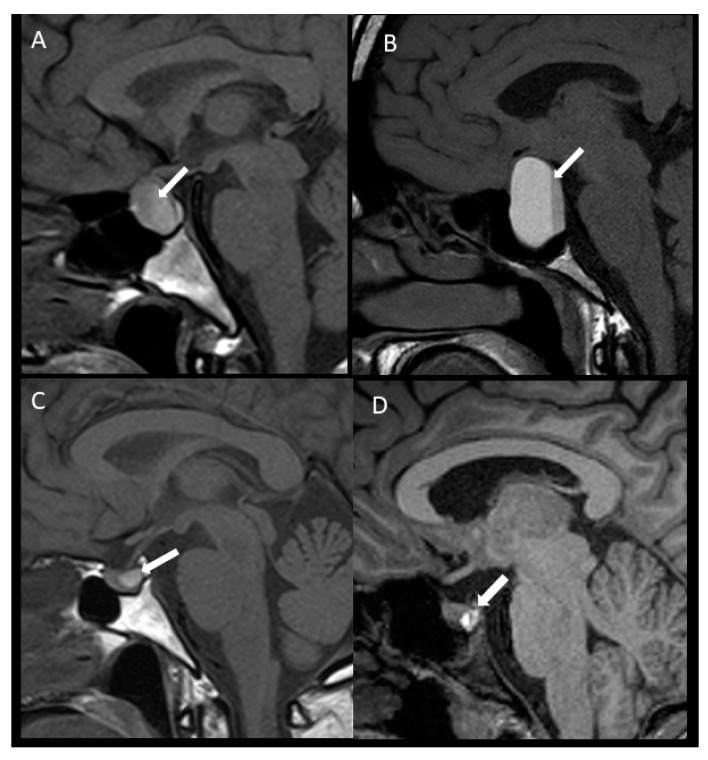
Pituitary apoplexy in macroadenoma (above) and microadenomas (below). Apoplexy in macroadenoma with subclinical presentation (**A**) and with acute clinical presentation (**B**), sagittal unenhanced T1WI, the intrasellar mass showed posterior hyperintensity that forms a level related to acute/subacute bleeding (arrow). Apoplexy in microadenoma with subclinical presentation (**C**) and with acute symptoms (**D**), sagittal unenhanced T1WI showed a microadenoma with hyperintensity related to bleeding (arrow). T1WI: T1-weighted image.

**Figure 2 jcm-11-07288-f002:**
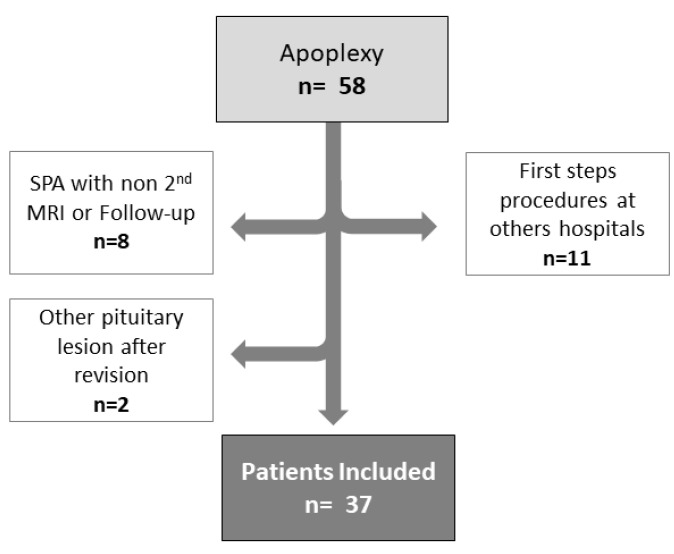
Flow chart of the study.

**Figure 3 jcm-11-07288-f003:**
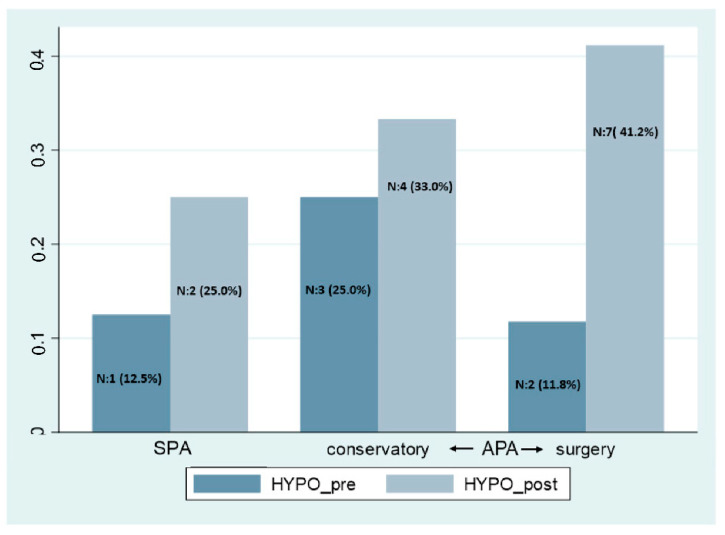
Hypopituitarism at apoplexy diagnosis and the follow-up. SPA: subclinical pituitary apoplexy, APA: acute pituitary apoplexy, Hypo: hypopituitarism.

**Figure 4 jcm-11-07288-f004:**
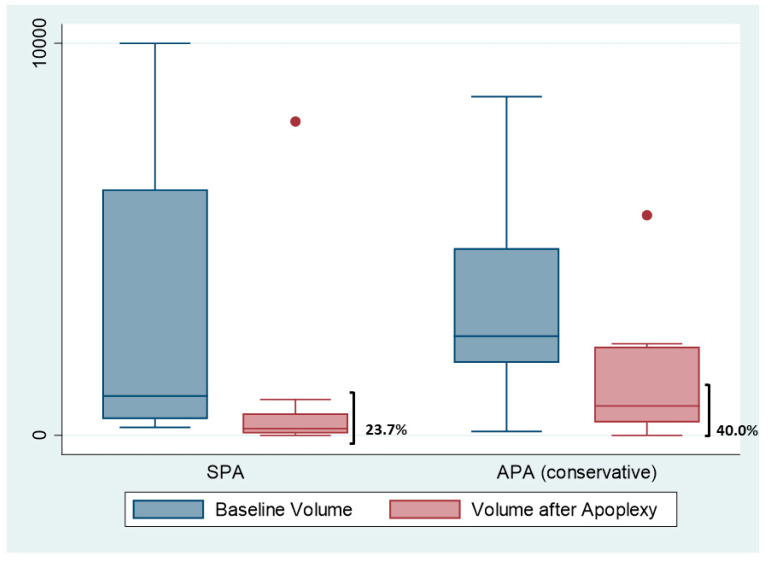
Initial volume and after 12 months of apoplexy. The figure shows the boxplots of the tumor’ volume reduction in subclinical apoplexy (SCA) and acute apoplexy (AA) treated conservatively. Dots are outliers.

**Figure 5 jcm-11-07288-f005:**
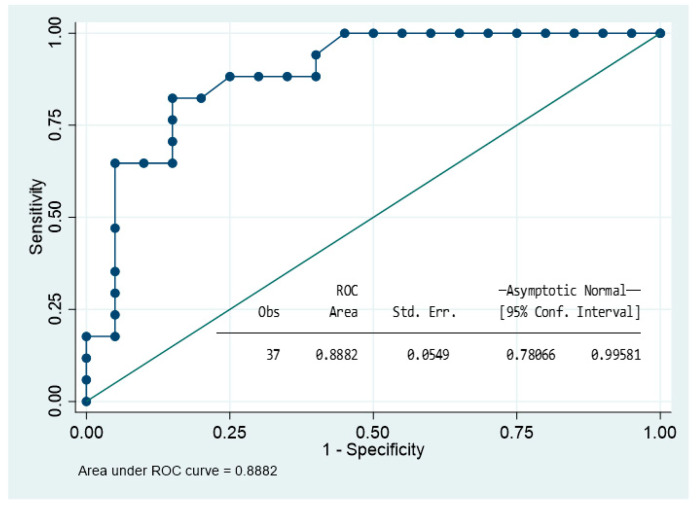
Roc curve showing the ability of the model to predict surgery in acute apoplexy group.

**Table 1 jcm-11-07288-t001:** Baseline demographic, clinical history, and risk factors of the different types of pituitary apoplexy.

	Total N = 37	Subclinical Apoplexy *n* = 8	Acute Apoplexy *n* = 29	*p*-Value
Men (*n*, %)	17 (46.0)	2 (25.0)	15 (51.7)	0.17
Age, years (Median; IQR)	47.7 (24.2)	42.2 (31.8)	51.2 (20.8)	0.25
BMI, Kg/m^2^ (Median; IQR)	28.5 (7.2)	24.7 (6.5)	27.7 (5.7)	0.11
T2DM (*n*, *%)*	11 (29.7)	0	11 (37.9)	0.04
HT (*n*, *%)*	10 (27.0)	1 (12.5)	9 (31.0)	0.40
DLP (*n*, *%)*	10 (27.0)	0	10 (34.4)	0.05
Tabaquisme (*n*, *%)*	5 (13.0)	2 (25.0)	3 (10.3)	0.29
Previous surgery (*n*, *%)*	5 (13.5)	2 (25.0)	3 (10.3)	0.29
Anticoagulant (*n*, *%)*	6 (16.2)	1(12.5)	5 (17.2)	0.61
Antiplatelet (*n*, *%)*	3(8.11)	0	3 (10.3)	0.47
CVD (*n*, *%)*	5 (13.5)	1 (12.5)	4 (13.8)	0.70
Previous DA (*n*, *%)*	5 (13.5)	2 (25.0)	3 (10.3)	0.29
Hypopituitarism (*n*, %)	6 (16.2)	1 (12.5)	5 (17.2)	0.61
Follow-up, months(Median; IQR)	23.0 (42.0)	33.9 (20.1)	22.9 (42.3)	0.56

BMI: body mass index, T2DM: Type 2 Diabetes Mellitus, HT: hypertension, DLP: dyslipemia, CVD: cardiovascular disease, TV: tumor’ volume MT mucosal thickening *p*-values: Fisher’s exact test for categorical variables and Mann–Whitney *U*-test for continuous variables, IQR: interquartile range.

**Table 2 jcm-11-07288-t002:** Radiological characteristics of the different types of pituitary apoplexy.

	Total N = 37	Subclinical Apoplexy *n* = 8	Acute Apoplexy *n* = 29	*p*-Value
Previous adenoma (*n*, *%)*	13 (35.1)	4 (50.0)	9 (31.0)	0.34
Macro	34 (91.9)	6 (75.0)	28 (96.6)	0.11
Diameter max. mm^3^ (Median; IQR)	25.0 (14.0)	13.5 (12.5)	26.0 (9.0)	0.01
TV mm^3^ (Median; IQR)	5049 (7049)	650 (879)	5589 (8562)	0.01
T1 hyperintense (*n*, *%)*	26 (70.3)	8 (100)	18 (62.1)	0.04
T2 hyperintense (*n*, *%)*	12 (32.4)	4 (50.0)	8 (27.6)	0.22
Radiological Hemorrhage	6 (16.2)	3 (37.5)	3 (10.3)	0.14
Radiological Infarction	3 (8.1)	0	3(10.3)	0.14
Mixed radiological Hemorrhage/Infarction	28 (75.7)	5 (62.5)	23(79.3)	0.14
Sphenoid sinus MT (*n*, *%)*	11 (29.7)	1 (12.5)	10 (34.4)	0.23
Ring enhancement (*n*, *%)*	11 (30.6)	2 (25.0)	9 (31.0)	0.64
Extrasellar (*n*, *%)*	25 (67.6)	2 (25.0)	25 (86.2)	0.01
Chiasm compression (*n*, *%)*	25 (67.6)	2 (25.0)	23 (79.3)	0.01
Sinus invasion (*n*, *%)*	28 (75.7)	3 (37.5)	25 (86.2)	0.01

Macro: macroadenoma, Micro: microadenoma, NFPA: non-functioning pituitary adenoma, PRL: prolactinoma, GH: acromegaly, max: maximum, MT: mucosal thickening, IQR: interquartile range. *p*-values: Fisher’s exact test for categorical variables and Mann–Whitney *U*-test for continuous variables,

**Table 3 jcm-11-07288-t003:** Differences between conservatory vs. surgical approach in acute apoplexy group.

	Total N = 29	Conservatory *n* = 12	Surgery *n* = 17	*p*-Value
Headache (*n*, %)	27 (93.1)	10 (83.3)	17 (100)	0.06
Nausea and vomiting (*n*, %)	10 (34.5)	2 (16.7)	8 (47.0)	0.10
Visual impairment (*n*, *%)*	21 (72.4)	7 (58.3)	14 (82.4)	0.16
Visual acuity (*n*, *%)*	13 (44.8)	3 (25.0)	10 (58.8)	0.08
Photophobia (*n*, *%)*	9 (31.0)	1(8.3)	8 (47.0)	0.03
Ocular paresis (*n*, *%)*	9 (31.0)	3 (25.0)	6 (35.1)	0.43
Confusion (*n*, *%)*	4 (13.8)	1(8.3)	3 (17.7)	0.44
Altered mental state (*n*, *%)*	1 (3.5)	0	1 (5.88)	--
Hyponatremia (*n*, *%)*	7 (24.7)	4 (33.3)	3 (17.7)	0.30
Hypopituitarism (*n*%) *	5 (17.2)	3 (25.0)	2 (11.8)	0.33
Previous adenoma (*n*, *%)*	10 (34.5)	7 (58.3)	3 (17.7)	0.07
Diameter max. mm^3^ (Median; IQR)	26.0 (9.0)	20.5 (11.5)	30.0 (9.0)	0.03
TV mm^3^ (Median; IQR)	5589 (8562)	2066 (5768)	8505 (8242)	0.00
T1 hyperintense (*n*, *%)*	18 (62.1)	9 (75.0)	9(52.9)	0.21
T2 hyperintense (*n*, *%)*	8 (27.6)	2 (16.7)	6 (35.3)	0.29
Sphenoid sinus MT (*n*, *%)*	10 (34.5)	3 (25.0)	7 (41.2)	0.34
Ring enhancement (*n*, *%)*	9(31.0)	5 (41.7)	4 (23.5)	0.58
Extrasellar (*n*, *%)*	23 (79.3)	8 (66.7)	15 (88.2)	0.17
Chiasm compression (*n*, *%)*	23 (79.3)	7 (58.3)	16 (94.1)	0.03
Sinus invasion (*n*, *%)*	25 (86.2)	10 (83.3)	15 (88.2)	0.56
Dexamethasone *n*: dose (mg) [Median; IQR]	9 (12.0; 4.0)	5 (12.0; 4.0)	4 (14.0; 12.0)	0.27
Hydrocortisone *n*: dose (mg) [Median; IQR]	17 (150.0; 200.0)	6 (135.0; 125.0)	9 (200.0; 150.0)	0.32

TV: tumor volume, *p*-values: Fisher’s exact test for categorical variables and Mann–Whitney U-test for continuous variable. * N = 28 One patient died in emergency room after emergency surgery, and hormonal state was not evaluated. IQR: interquartile range, MT: mucosal thickening.

**Table 4 jcm-11-07288-t004:** Multivariable analysis to predict surgery.

	OR	OR 95% CI	*p*
NFPA	29.36	1.86 462.33	0.016
Max. Diameter	1.11	1.01 1.20	0.036
Hyper-T1-WI	0.08	0.01 1–14	0.063

NFPA: non-functioning pituitary adenoma. Max: maximum; T1-WI: T1 weighted imaging on MRI OR: odds ratio, CI: confidence interval.

## Data Availability

The datasets generated during and/or analysed during the current study are not publicly available but are available from the corresponding author at any reasonable request.

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
