# Peer review of "Risk Factors, Radiological and Clinical Outcomes in Subclinical and Clinical Pituitary Apoplexy"

_jcm, 2022, doi:10.3390/jcm11247288_

Round 1

Reviewer 1 Report

The authors analyzed 37 patients with pituitary apoplexy from 2010 to 2021, including 29 acute symptomatic apoplexy (APA), 8 clinically asymptomatic or subclinical apoplexy (SPA). The clinical characteristics and treatment outcome were analyzed.

In the section of 3.5. Prediction model for surgery, the probability to required surgery included: to have a NFPA, hyperintensity on T1-WI, and maximum diameter on MRI. As a whole it were (24, 64.9%) NFPAs, (10, 27.0%) prolactinomas and (3, 8.1%) GH secreting adenomas. Obviously, for such a conclusion, there is a bias of insufficient data. On the one hand, the number of cases of each PitNET subtype is not enough, and more importantly, the total number of PitNET cases is not enough.

Another aspect of the insufficient number of cases affecting the statistical results is that in Table 2, some factors cannot be counted due to the too small number of cases. Clinically, the number of cases of SPA should be more, which is also a doubt about the results of article.

There are minor concerns:

1,   Figure 1 should be presented in the section of Patients and Methods;

2,   In the figure 1 C and D, considering apoplexy, require more imaging support.

3,   Page 4 line 119,subtitle should be Inclusion criteria.

4,   Line 167, how many pituitary tumor patients?

5,   Line181-183,Type 2 Diabetes Mellitus (T2DM) and dyslipemia were more frequent in APA group. This result seems to be difficult to the clinical interpretation, is it related to the small number of SPA cases?

6,   In the Table 2, the incidence of sinus invasion in APA is very high, accounting at 86.2%, and the authors need to explain the definition of sinus invasion.

7,   In non-surgical group, the pituitary tumour shrinkage is clinically relevant after one year of PA. I don't know if the authors have any MR data for 3 months and 6 months after apoplexy?

Author Response

The authors analyzed 37 patients with pituitary apoplexy from 2010 to 2021, including 29 acute symptomatic apoplexy (APA), 8 clinically asymptomatic or subclinical apoplexy (SPA). The clinical characteristics and treatment outcome were analyzed.

Answer: Many thanks for the revision process of our manuscript. We have addressed all your comments, which has significantly improved the quality of our paper.

In the section of 3.5. Prediction model for surgery, the probability to required surgery included: to have a NFPA, hyperintensity on T1-WI, and maximum diameter on MRI. As a whole it were (24, 64.9%) NFPAs, (10, 27.0%) prolactinomas and (3, 8.1%) GH secreting adenomas. Obviously, for such a conclusion, there is a bias of insufficient data. On the one hand, the number of cases of each PitNET subtype is not enough, and more importantly, the total number of PitNET cases is not enough.

Answer: We agree with the reviewer that larger samples would give stronger support to our results and we have added your concern as limiting factors of our study page 14 lines 381-384.  However, we would like to comment the following:

First, from a statistical point of view, although under debate, ten events per independent variable is the minimal criterion for sample size considerations in logistic regression analysis. Thus, we accomplish with the rule and the model was well fitted according to Akaike’s and Bayesian requirements. In addition, the Likelihood ratio compared with the null model was <0.01

Second, PitNET are rare diseases and  in this setting apoplexy  is  a really infrequent event. Our database registered 529 PitNET from 2010 up to the study close and the prevalence of apoplexy in each tumor type in our series was [24/121, (19.8%), for NFPA], [10/189, (5.2%) for prolactinomas], [3/74 (4.1%) for acromegaly]. No patients in our database with Cushing disease had PA. Thus, the sample seems to be representative. We have also added this useful information in page 6 lines 178-179 and page 7 lines 208-211

Another aspect of the insufficient number of cases affecting the statistical results is that in Table 2, some factors cannot be counted due to the too small number of cases. Clinically, the number of cases of SPA should be more, which is also a doubt about the results of article.

Answer: Thanks for your comment. We agree with the referee that some variables as to have a micro or type of secretion cannot be tested for differences due to the low prevalence in our sample. We have deleted these variables from Table 2 of the revised manuscript. However, we would underline that , despite of having a limited sample, the statistical power was adequate to find differences between   SPA vs AA

There are minor concerns:

1,   Figure 1 should be presented in the section of Patients and Methods;

Answer:  Thanks for comment! We have moved it to Page 5 at the end of Patients and Methods section.

2,   In the figure 1 C and D, considering apoplexy, require more imaging support.

Answer:  We will replace the image by TIFF version to assure a better edition.

3,   Page 4 line 119,subtitle should be Inclusion criteria..

Answer:  Done!

4,   Line 167, how many pituitary tumor patients?

Answer:  Our database registered 529 PitNET up to the study close. We have added this information page 6 lines 178-179 of the revised manuscript. We have then included the prevalence of apoplexy in our series as commented above.

5,   Line 181-183,Type 2 Diabetes Mellitus (T2DM) and dyslipemia were more frequent in APA group. This result seems to be difficult to the clinical interpretation, is it related to the small number of SPA cases?

Answer: The referee is right in indicating this point, being a referral Hospital could imply a potential bias to more APA than SPA. We have included into the limiting factor of our paper,  page 14 lines 381-384. From a clinical point of view, the classic cardiovascular risk factors such as diabetes, hypertension, and dyslipidemia, are not well-established risk factors for pituitary apoplexy. The presence of diabetic microangiopathy could favour pituitary apoplexy and, theoretically, this could be more frequent in larger tumours. However, as we commented in the discussion (page 13 lines 307-308), although plausible, some works have not found this association.  

6,   In the Table 2, the incidence of sinus invasion in APA is very high, accounting at 86.2%, and the authors need to explain the definition of sinus invasion.

Answer: Parasellar tumour involvement was assessed by Knosp classification, we have added the reference in page 4 line 140 of the revised manuscript (highlighted in yellow)

7,   In non-surgical group, the pituitary tumour shrinkage is clinically relevant after one year of PA. I don't know if the authors have any MR data for 3 months and 6 months after apoplexy?

Answer: We have data of most patients with acute apoplexy at 3-6 months, but we don’t  have this information  in the  case of subacute apoplexy. Therefore, for comparing both groups we have used the  MRI performed at one year of follow-up.

Reviewer 2 Report

Revision

Risk factors, radiological and clinical outcomes in subclinical and clinical pituitary apoplexy

Dear Editors and authors,

Thank you for the opportunity to review this article titled “Risk factors, radiological and clinical outcomes in subclinical and clinical pituitary apoplexy”. This article is really interesting, at this point in time, this study has a few limitations and considerable modifications need to be done to improve its quality. Some main areas of concerns are described below:

According to instructions for authors, The abstract should be a total of about 200 words maximum. This version, 280 words.

Please add a full stop to the last sentence: “Consequently, surgery indication in NFPA should be delayed and reassess if patients remain symptomatic”

Please. Delete the numbers into keywords.

Please, check the text, there is an inconsistency in the use of double spaces. Please a sentence cannot be a paragraph.

Is the frequency of SPA around the world or in Spain?

Please describe this study according its design. I recommended consult the following website: https://www.equator-network.org/

I suggest the authors add the Youden Index, it is of great interest

  great interest

Author Response

Thank you for the opportunity to review this article titled “Risk factors, radiological and clinical outcomes in subclinical and clinical pituitary apoplexy”. This article is really interesting, at this point in time, this study has a few limitations and considerable modifications need to be done to improve its quality. Some main areas of concerns are described below:

Answer: Many thanks for the revision process and your kind comments on our manuscript.

According to instructions for authors, The abstract should be a total of about 200 words maximum. This version, 280 words.

Answer: Thanks for comment! We have shortened the abstract to accomplish with the journal instructions.

Please add a full stop to the last sentence: “Consequently, surgery indication in NFPA should be delayed and reassess if patients remain symptomatic”

Answer: Thank you!

Please. Delete the numbers into keywords.

Answer: Done!

Please, check the text, there is an inconsistency in the use of double spaces. Please a sentence cannot be a paragraph.

Answer: We have carefully revised the text as suggested.

Is the frequency of SPA around the world or in Spain?

Answer:  The percentage is refereed to different studies world-wide.   We have pointed it in page 1 line 42 of the revised manuscript

Please describe this study according its design. I recommended consult the following website: https://www.equator-network.org/

Answer: Many thanks for your useful suggestion. We have added that we followed the strobe rules for cross sectional studies page 3 line 96 and we have placed a new reference number 19 highlighted in yellow in the revised manuscript.

I suggest the authors add the Youden Index, it is of great interest

Answer: we used STATA “estat classification” command to know the discrimination power of the model. The model Correctly classified 83.78% and we feel that this index is easier to understand for a clinician than Youden Index (-1 to 1), being close to one the best.  Nevertheless, following your recommendation, we also calculated Youden Index for the model, which was 0.687 and this information has been added to the revised manuscript Page 11 lines 277-278.